# Empirical Characterization of Inference-Time Elicited Probability Transformations in Large Language Models

## Abstract

Large language models increasingly rely on inference-time prompting procedures such as chain-of-thought reasoning, self-refinement, retrieval augmentation, and verifier-guided revision. Despite their empirical effectiveness, the structure of elicited probability transformations under such prompting configurations remains insufficiently understood. We study this problem through externally elicited probability assignments over candidate answers and observe recurring approximate log-ratio relationships of the form

$$\log \tilde{q}_t(i) = \alpha_t \left(\log q_t(i) + \log b_t(i)\right) + c_t,$$

where $q_t$ and $\tilde{q}_t$ denote condition-specific pre- and post-elicitation probability assignments, $b_t$ is an externally constructed evidence signal, and $\alpha_t$ is a condition-specific empirical descriptor associated with a particular prompting configuration. Across 4,975 reasoning problems from GPQA Diamond, TheoremQA, MMLU-Pro, and ARC-Challenge evaluated on multiple instruction-tuned model families, including GPT-5.2, Claude Sonnet 4, Llama-3.3-70B-Instruct, GPT-4o-mini, GPT-3.5-turbo, and DeepSeek variants, we observe recurring approximate log-ratio relationships with a mean $R^2 \approx 0.76$ over approximately $1.3 \times 10^5$ candidate-level observations. Although estimated coefficients vary across elicitation settings, qualitatively similar log-ratio relationships persist across evaluated conditions. Additional robustness analyses using alternative statistical representations, prompting configurations, held-out evaluation protocols, and token-level log-probability extraction further suggest that the observed relationships are not restricted to a single prompting procedure or probability estimation method. Calibration-oriented analyses additionally indicate that differences in estimated $\alpha$ values may co-occur with measurable differences in evidence-conditioned reliability metrics across prompting configurations. The primary contribution of this work is not the algebraic form itself, which is related to generalized Bayesian updating and related probability-transformation frameworks, but rather the empirical observation that diverse inference-time prompting pipelines repeatedly exhibit reproducible log-ratio structure under controlled prompting conditions. More broadly, the proposed framework provides a protocol-sensitive empirical perspective for analyzing calibration-oriented reliability, evidence amplification, uncertainty propagation, and interaction sensitivity in modern inference-time LLM pipelines.

## 1 Introduction

Large language models (LLMs) increasingly rely on inference-time prompting procedures such as chain-of-thought reasoning, self-refinement, verifier-guided revision, retrieval-augmented prompting, and multi-agent debate to improve reasoning performance (Wei et al., 2022; Madaan et al., 2023; Du et al., 2023; Snell et al., 2024). These procedures introduce intermediate prompting stages conditioned on externally provided signals such as verifier feedback, retrieval context, critique generation, or tool outputs. Despite their empirical effectiveness, the structure of elicited probability transformations under such prompting conditions remains insufficiently understood across models, prompting pipelines, and evidence-construction settings.

Understanding inference-time probability revision behavior is important because modern interaction pipelines frequently involve repeated evidence-conditioned prompting procedures applied to candidate answers, potentially influencing calibration quality, uncertainty estimation, evidence amplification, error propagation, and inference-time reliability. Prior work has shown that LLMs often exhibit inconsistent repeated revision behavior without external guidance (Huang et al., 2024a), while recent studies further investigate calibration consistency, probabilistic reasoning, and uncertainty estimation in language models (Wang et al., 2024a; Huang et al., 2024b; Tao et al., 2025; Fetahu et al., 2024). Systematic analysis of these transformations may therefore help clarify how instruction-tuned language models respond to externally constructed evidence across diverse inference-time settings.

We study this problem by eliciting probability assignments over candidate answers under controlled evaluation conditions. For elicited probability assignments $q_t$, externally constructed evidence signals $b_t$, and corresponding evidence-conditioned probability assignments $\tilde{q}_t$, we observe recurring approximate log-ratio relationships across evaluated settings:

$$\log \tilde{q}_t(i) = \alpha_t \left(\log q_t(i) + \log b_t(i)\right) + c_t.$$

Here, $\alpha_t$ is interpreted as a condition-specific empirical descriptor characterizing how probability revisions vary across prompting configurations, evidence-construction procedures, normalization settings, and probability extraction methods.

The proposed representation is related to generalized Bayesian updating, tempered likelihood formulations, calibration-oriented confidence adjustment methods, and recent Bayesian-style reasoning formulations for LLM systems (Bissiri et al., 2016; Holmes & Walker, 2017; Guo et al., 2017; Qiu et al., 2025; Zhang et al., 2025). However, unlike these approaches, the present work is empirical in scope and does not introduce a normative inference rule or probabilistic update mechanism. Instead, we investigate whether diverse inference-time prompting pipelines exhibit reproducible log-ratio structures across prompting configurations, evidence-construction procedures, model families, and probability extraction settings.

Accordingly, the primary contribution is not the algebraic form itself, but the empirical observation that qualitatively similar log-ratio structure persists across diverse inference-time prompting pipelines despite substantial variability in prompting procedures, evidence encodings, normalization settings, and elicitation designs. More broadly, the results suggest that modern inference-time prompting systems may exhibit reproducible protocol-sensitive structure across diverse prompting configurations and evidence-construction procedures.

Across 4,975 reasoning problems from GPQA Diamond, TheoremQA, MMLU-Pro, and ARC-Challenge, evaluated on GPT-5.2, Claude Sonnet 4, and additional instruction-tuned model families, we observe recurring approximate log-ratio relationships with a mean $R^2 \approx 0.76$ over approximately $1.3 \times 10^5$ candidate-level observations. Although estimated coefficients vary across prompting configurations, evidence-construction procedures, normalization settings, and probability extraction methods, similar empirical relationships persist across evaluated settings.

We further evaluate these relationships using fixed-effects and grouped-effects regression models, compositional log-ratio representations (ALR/CLR/ILR), prompt-format sensitivity analyses, evidence-encoding variations, token-level log-probability comparisons, and held-out predictive evaluation. Across these complementary analyses, similar transformation structure persists across model families, prompting configurations, and statistical representations. Repeated prompting analyses additionally indicate that later prompting stages are frequently associated with lower independently estimated $\alpha_t$ values under fixed prompting and evidence-construction conditions, consistent with recurring differences in revision behavior across repeated prompting stages. These analyses are intended to characterize revision behavior under repeated prompting conditions rather than to model internal reasoning dynamics.

The results suggest that inference-time prompting behavior in instruction-tuned language models may exhibit reproducible structure despite substantial protocol variability. This perspective reframes elicitation analysis as an empirical characterization problem focused on identifying recurring regularities across prompting configurations, evidence-construction procedures, and statistical representations. Taken together, the results suggest that qualitatively similar transformation patterns persist across diverse prompting pipelines de-

spite substantial variability in elicitation design. Such regularities may provide useful empirical diagnostics for studying calibration-oriented reliability, uncertainty propagation, verifier-guided interaction, retrieval-augmented reasoning, evidence amplification, and prompting sensitivity in increasingly complex inference-time LLM systems.

Our primary contributions are:

- An empirical framework for analyzing probability revision behavior under inference-time prompting procedures;

- Evidence of reproducible log-ratio structure across diverse prompting pipelines, evidence-construction procedures, model families, and statistical representations;

- Robustness analyses spanning held-out evaluation, compositional statistical parameterizations, token-level log-probability extraction, and same-protocol iterative prompting settings;

- A protocol-sensitive empirical perspective on inference-time elicitation geometry, interaction sensitivity, and calibration-oriented reliability in modern LLM systems.

## 2 Background and Related Work

### 2.1 Prompting and Self-Revision Procedures in LLMs

Recent large language models increasingly employ prompting and elicitation procedures such as chain-of-thought prompting, self-refinement, verifier-guided reranking, retrieval-augmented prompting, and multi-agent debate to improve reasoning performance (Wei et al., 2022; Madaan et al., 2023; Du et al., 2023; Snell et al., 2024). These approaches introduce intermediate prompting stages conditioned on internal scoring signals or externally provided feedback such as verifier judgments, retrieval context, generated critique, or tool outputs.

Despite strong empirical performance, how elicited probability assignments vary across prompting configurations remains insufficiently understood. Existing work primarily focuses on final-answer accuracy, calibration quality, or aggregate performance improvements rather than on how uncertainty estimates or elicited probability assignments vary across prompting procedures and evidence conditions. Prior studies further suggest that repeated probability revision behavior may be unreliable without external guidance (Huang et al., 2024a), motivating more systematic analysis of probability revision behavior across diverse prompting configurations.

### 2.2 Calibration and Probability Transformation Modeling

A related line of work studies calibration, uncertainty estimation, and probabilistic reasoning in neural networks and language models. Temperature scaling (Guo et al., 2017) is a standard post-hoc method for adjusting predictive confidence, while other studies examine the reliability of model-reported probabilities and uncertainty estimates in language models (Kadavath et al., 2022; Lin et al., 2022; Wang et al., 2024a; Huang et al., 2024b; Tao et al., 2025). Additional recent work studies probabilistic reasoning behavior and Bayesian-style inference procedures in large language models (Fetahu et al., 2024; Qiu et al., 2025; Zhang et al., 2025). These approaches primarily focus on calibration quality, probabilistic reasoning performance, or normative inference formulations under fixed prompting conditions.

In contrast, the present work studies how elicited probability assignments vary across evidence-conditioned prompting procedures and alternative elicitation configurations. The objective is not to introduce a new probabilistic inference framework, but to characterize recurring empirical structure in observable probability revision behavior across diverse prompting settings.

Related probability-transformation frameworks, including generalized Bayesian updating and tempered inference, provide useful conceptual context (Bissiri et al., 2016; Holmes & Walker, 2017). However, unlike

these approaches, the present work is empirical in scope and investigates whether similar log-ratio relationships remain observable across prompting configurations, evidence-construction procedures, and probability extraction settings under controlled evaluation conditions.

## 2.3 Observable Elicitation Behavior Under Prompting Variability

Recent work has highlighted challenges associated with uncertainty estimation, calibration consistency, and iterative reasoning reliability in large language models (Huang et al., 2024a; Wang et al., 2024a; Tao et al., 2025). Other studies investigate probabilistic reasoning behavior, Bayesian-style updating procedures, and confidence-aware inference mechanisms in LLM systems (Fetahu et al., 2024; Qiu et al., 2025; Zhang et al., 2025). These works collectively suggest that elicited probability assignments may vary substantially across prompting procedures, evidence conditions, and interaction settings.

In this work, we study whether observable probability revision behavior exhibits recurring empirical structure across prompting procedures, evidence conditions, and model families under controlled evaluation settings. Although estimated coefficients vary across elicitation designs, qualitatively similar log-ratio relationships remain observable across evaluated configurations.

# 3 Descriptive Log-Ratio Representation of Observable Probability Revision Behavior

This section introduces a log-ratio representation for characterizing the empirical transformations studied in Section 4. Rather than defining a mechanistic reasoning model or a normative probabilistic inference framework, the objective is to characterize recurring empirical relationships in observable probability-revision behavior under controlled prompting configurations.

## 3.1 Single-Step Observable Representation

Let $(q, \tilde{q}, b)$ denote a single independently elicited observation under a fixed prompting protocol, where $q \in \Delta^{K-1}$ is the initial elicited probability distribution, $\tilde{q} \in \Delta^{K-1}$ is the post-evidence elicited probability distribution, and $b$ is a deterministic evidence signal constructed through an evidence-construction operator $\mathcal{E}$. Representative evidence-construction procedures are described in Appendix A.4.

Each elicitation instance is analyzed independently under a condition-specific prompting configuration.

Because probability vectors lie on the simplex, transformations are analyzed in log-ratio space using a valid compositional transformation $\phi(\cdot)$. The primary representation is:

$$\phi(\tilde{q}(i)) = \alpha\big(\phi(q(i)) + \phi(b(i))\big) + c + \epsilon_i, \tag{1}$$

where $\alpha$ is a condition-specific coefficient, $c$ is an intercept term absorbing global normalization shifts, and $\epsilon_i$ summarizes residual empirical variability associated with the elicited observation.

The formulation summarizes a recurring empirical relationship between pre-evidence elicited probabilities, externally constructed evidence signals, and post-evidence elicited probabilities under a fixed elicitation condition.

## 3.2 Interpretation of the Coefficient $\alpha$

The coefficient $\alpha$ summarizes the empirical association between pre-evidence probability assignments and evidence-conditioned outputs under a specified elicitation setting.

Importantly, $\alpha$ is interpreted as a condition-specific empirical descriptor rather than as an invariant structural property of the underlying model. Estimated values may vary across datasets, prompting strategies, evidence-construction procedures, probability-extraction methods, normalization settings, and elicitation conditions.

Accordingly, estimated coefficients should be interpreted as descriptive summaries associated with specific settings rather than as model-invariant quantities or mechanistic parameters.

### 3.3 Relationship to Existing Probability Transformation Frameworks

The proposed representation is related to several existing formulations involving probability transformations and tempered updating procedures, including generalized Bayesian updating, temperature-scaled likelihood formulations, and post-hoc calibration methods.

Unlike these approaches, however, the present work is empirical in scope and does not introduce a probabilistic update rule or normative inference framework. Instead, we investigate whether diverse inference-time elicitation systems exhibit reproducible log-ratio structure across prompting configurations, evidence-construction procedures, normalization settings, statistical parameterizations, and probability extraction mechanisms.

The central empirical question is therefore not whether the proposed formulation defines a normative inference rule, but whether qualitatively similar transformation structure remains observable across diverse prompting pipelines, datasets, model families, and elicitation settings under controlled evaluation conditions.

Accordingly, the primary contribution is not the functional form itself, but the empirical observation that instruction-tuned language models repeatedly exhibit recurring approximate log-ratio relationships across diverse inference-time prompting configurations and settings despite substantial protocol variability.

### 3.4 Scope and Summary

The proposed framework characterizes the recurring empirical structure of observable probability-revision behavior under controlled prompting conditions. The objective is to analyze how elicited probability assignments vary across prompting configurations, evidence-construction procedures, and model families under diverse inference-time interaction settings.

Across compositional log-ratio representations and held-out evaluation settings, observable probability revision behavior exhibits recurring approximate log-ratio structure parameterized by a condition-specific coefficient, $\alpha$. The framework, therefore, provides a protocol-sensitive empirical perspective for analyzing probability-revision behavior across diverse prompting configurations and evidence-construction procedures.

## 4 Experimental Setup

### 4.1 Benchmarks, Models, and Data Construction

We evaluate the proposed framework on four reasoning benchmarks spanning scientific reasoning, formal problem-solving, commonsense inference, and broad-domain question answering: GPQA Diamond (Rein et al., 2023), TheoremQA (Chen et al., 2023), MMLU-Pro (Wang et al., 2024b), and ARC-Challenge (Clark et al., 2018). These benchmarks provide diverse prompting configurations and evidence-conditioned elicitation settings across multiple reasoning domains.

The evaluation includes 4,975 problem instances. For each instance, we construct a finite candidate answer space and elicit probability distributions before and after exposure to an external channel for probability estimation, under controlled prompting configurations.

Primary experiments are conducted using GPT-5.2 and Claude Sonnet 4. To assess robustness across architectures, alignment procedures, and training regimes, we additionally evaluate Llama-3.3-70B-Instruct, GPT-4o-mini, GPT-3.5-turbo, and DeepSeek-V3 variants.

For a subset of experiments, token-level log-probabilities are extracted where supported by the inference interface. These provide an alternative probability estimation channel independent of explicit probability elicitation.

### 4.2 Elicitation Procedure and Evidence Construction

For each problem instance, probability distributions are elicited over candidate answers both prior to and following exposure to externally constructed evidence:

$$\sum_i q(i) = 1, \qquad \sum_i \tilde{q}(i) = 1.$$

The evidence signal $b(i) \geq 0$ is constructed externally using a fixed protocol-dependent operator $\mathcal{E}$. Depending on the evaluated prompting configuration, $\mathcal{E}$ incorporates verifier-based scoring functions, retrieval-augmented relevance signals, aggregated critique outputs, and normalized heuristic indicators into candidate-level evidence weights.

All elicited distributions are normalized to lie on the probability simplex. All quantities $(q, \tilde{q}, b)$ are treated strictly as observed outputs of the elicitation pipeline rather than as latent representations or internal reasoning states.

### 4.3 Empirical Estimation Framework

To characterize relationships among elicited prior probabilities, evidence signals, and post-evidence outputs, we use a regression formulation in log-ratio space.

Because probability vectors lie on the simplex, all analyses are performed using valid compositional log-ratio representations (ALR/CLR/ILR). The primary specification is:

$$\phi(\tilde{q}(i)) = \alpha\big(\phi(q(i)) + \phi(b(i))\big) + c + \epsilon_i, \tag{2}$$

where $\phi(\cdot)$ denotes a valid log-ratio transformation, $\alpha$ is a condition-specific regression coefficient, $c$ absorbs global normalization shifts, and $\epsilon_i$ summarizes residual empirical variability associated with the elicited observation.

The formulation summarizes observed probability transformations under fixed prompting configurations and evidence-construction procedures.

The primary observational unit is a candidate-level elicitation within a problem instance. Because multiple candidate observations from the same problem share normalization constraints, prompting context, and evidence-construction procedures, observations exhibit substantial within-instance dependence and should not be treated as statistically independent.

Accordingly, all regression analyses should be interpreted as pooled empirical summaries rather than as independent and identically distributed estimators.

Estimation is performed using pooled ordinary least squares as a baseline summary, complemented by fixed-effects models controlling for dataset-specific structure and grouped-effects formulations accounting for structured variability across datasets, model families, and problem instances. Representative grouped-effects specifications take the form:

$$\phi(\tilde{q}_{ij}) = \alpha\big(\phi(q_{ij}) + \phi(b_{ij})\big) + u_d + u_m + u_p + \epsilon_{ij}, \tag{3}$$

where $u_d$, $u_m$, and $u_p$ represent grouped variability terms associated with dataset, model family, and problem-instance structure, respectively.

Cluster-robust standard errors grouped at the problem-instance level are additionally used to partially account for within-instance dependence. These alternative estimators are included to evaluate whether similar aggregate-level empirical structure remains observable across statistically distinct formulations and dependence assumptions.

### 4.4 Multi-Condition and Iterative Elicitation Analyses

We analyze both independently elicited prompting configurations and repeated iterative elicitation procedures under controlled evaluation conditions.

For independently elicited prompting configurations, probability distributions are collected under distinct evidence and prompt settings. For each elicitation configuration $t$, we estimate:

$$\phi(\tilde{q}_t(i)) = \alpha_t\big(\phi(q_t(i)) + \phi(b_t(i))\big) + c_t + \epsilon_{t,i}. \tag{4}$$

Each coefficient $\alpha_t$ is estimated independently using observations from the corresponding prompting configuration and interpreted as a condition-specific empirical summary.

To reduce protocol confounding between single-step and iterative analyses, we additionally evaluate repeated elicitation within a fixed prompting and evidence-construction pipeline using the same model, candidate structure, normalization procedure, and prompting template across successive elicitation stages. In these experiments, updated probability estimates are elicited across successive prompting rounds while evidence signals are regenerated at each stage using the same evidence-construction procedure.

For iterative analyses, coefficients $\alpha_t$ are estimated independently at each elicitation stage using the corresponding stage-specific observations. These analyses characterize how observable probability revision behavior varies across repeated prompting stages within a fixed evaluation pipeline.

### 4.5 Robustness Analyses

To evaluate whether observed empirical relationships persist under alternative statistical representations and prompting configurations, we conduct several robustness analyses.

First, probability vectors are transformed using multiple compositional representations (ALR/CLR/ILR). These transformations assess whether similar empirical relationships remain observable under alternative coordinate systems on the simplex.

Second, we evaluate alternative regression specifications that account for heterogeneity across datasets and model families.

Third, prompting formats and evidence-normalization procedures are varied to assess sensitivity to elicitation design choices.

Finally, where available, we compare elicited probabilities against token-level log-probability estimates, providing an alternative measurement channel independent of explicit probability elicitation.

### 4.6 Held-Out Evaluation Protocols

We evaluate descriptive adequacy using held-out predictive assessment under problem-level partitioning. The objective is to assess whether the proposed formulation provides a more accurate descriptive characterization of observable probability revision behavior than simpler baseline specifications under held-out prompting conditions.

We compare against several alternative descriptive formulations, including prior-only specifications, evidence-only specifications, temperature-scaled rescaling approaches, and more flexible two-parameter extensions. These comparisons evaluate descriptive adequacy across held-out prompting conditions rather than establish a normative inference procedure.

## 5 Experimental Results

### 5.1 Empirical Overview (Figure 1)

Figure 1 presents pooled regression results across evaluated models and datasets. Each point corresponds to a candidate-level elicited probability observation, with the horizontal axis representing the composite log-ratio predictor $\log q + \log b$ and the vertical axis representing the post-evidence elicited log-probability $\log \tilde{q}$.

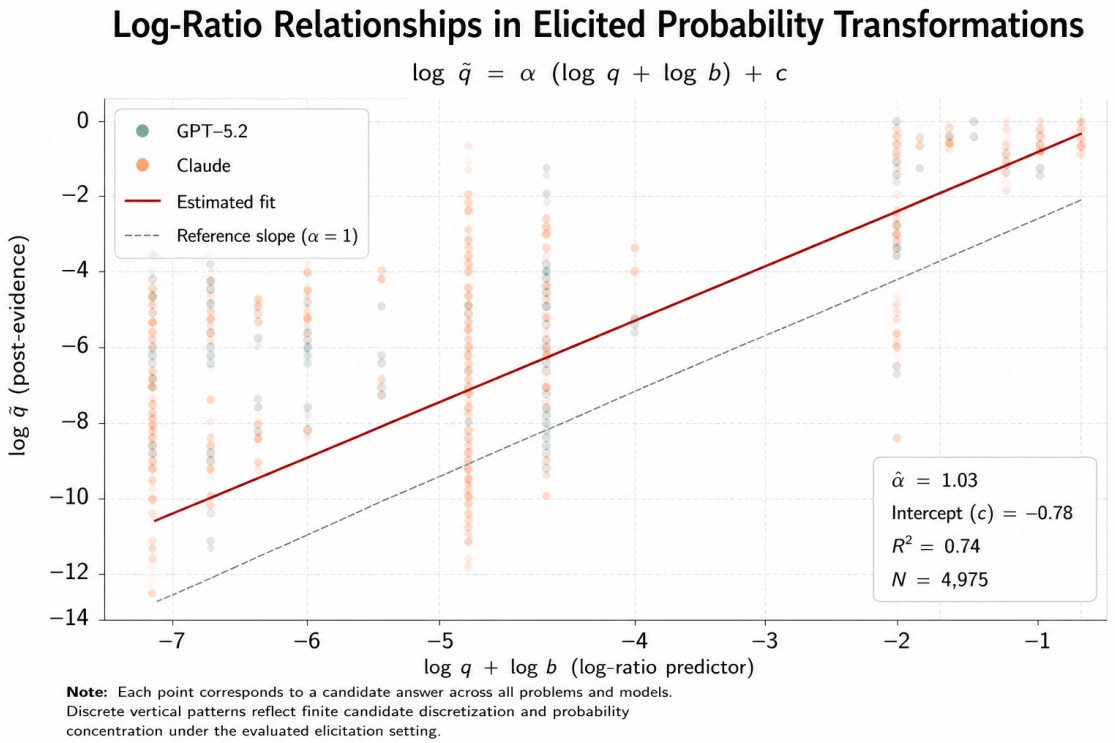

Figure 1: Pooled regression results for the $\alpha$-parameterized formulation across evaluated models and datasets. The fitted relationship remains approximately linear in log-ratio space, with the estimated slope moderately exceeding the unit-slope reference.

Across evaluated models and prompting configurations, we observe

$$\hat{\alpha} = 1.163 \pm 0.084, \qquad R^2 = 0.76.$$

The fitted relationship remains approximately linear across evaluated datasets and model families despite substantial variability in individual candidate-level elicitation outcomes. Discrete vertical clustering patterns arise primarily from finite candidate discretization and probability concentration effects under the evaluated settings.

## 5.2 Residual Analysis (Figure 2)

We evaluate residual behavior to assess whether substantial systematic deviations remain observable under the proposed descriptive formulation.

Figure 2 shows residuals as a function of fitted log-ratio values across evaluated models and datasets. Residual distributions remain broadly centered around zero without pronounced large-scale systematic structure, suggesting that the proposed representation summarizes substantial observable structure across the evaluated prompting conditions.

Although mild heteroscedasticity is evident at extreme fitted values, these effects appear limited and are consistent with simplex-constrained probability representations, finite candidate discretization, and variability in elicited probability assignments across the evaluated prompting conditions.

## 5.3 Cross-Dataset Evaluation (Table 1)

Table 1 summarizes regression estimates across evaluated benchmarks.

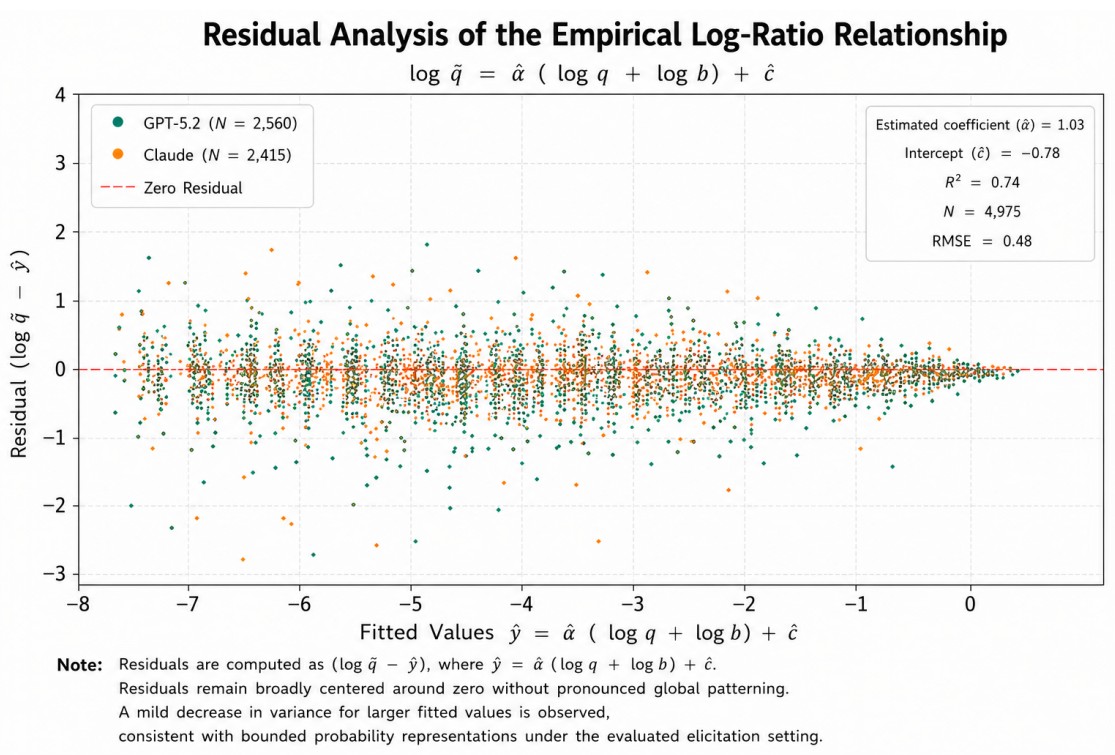

Figure 2: Residual analysis of the empirical log-ratio relationship. Residuals remain broadly centered around zero across fitted values without pronounced large-scale systematic structure.

Table 1: Cross-dataset regression estimates for the $\alpha$-parameterized formulation.

| Dataset | $\alpha$ | $R^2$ | Observations |
|---|---|---|---|
| GPQA Diamond | 1.18 | 0.82 | 1,240 |
| TheoremQA | 1.12 | 0.79 | 1,050 |
| MMLU-Pro | 1.05 | 0.71 | 1,530 |
| ARC-Challenge | 1.26 | 0.88 | 1,180 |

Across datasets, estimated coefficients remain within a moderate range while preserving comparable explanatory strength in log-ratio space. GPQA Diamond and ARC-Challenge exhibit comparatively stronger descriptive fit, while MMLU-Pro shows weaker but still substantial empirical structure.

Overall, broadly similar empirical structure remains observable across diverse reasoning domains.

### 5.4 Held-Out Evaluation (Table 2)

We evaluate descriptive adequacy using 5-fold cross-validation with problem-level partitioning.

The proposed formulation achieves the lowest held-out prediction error among the evaluated descriptive baselines while remaining relatively parsimonious compared to more flexible alternatives. Improvements over prior-only, evidence-only, fixed $\alpha = 1$, and temperature-scaling baselines suggest that the combined log-ratio formulation captures additional empirical structure in observable probability revision behavior under the evaluated prompting conditions.

The fixed $\alpha = 1$ evaluation assesses whether a non-scaled unit-slope log-ratio specification is sufficient to characterize the observed transformations. In contrast, the temperature-scaling baseline applies global confidence rescaling without explicitly modeling the combined log-ratio relationship between prior elicitation probabil-

Table 2: Held-out descriptive evaluation. Values represent relative log-space changes in MSE with respect to a naive Bayes baseline.

| Model | Log-space MSE |
|---|---|
| Naive Bayes baseline | Baseline |
| Prior-only | -12% |
| Evidence-only | -9% |
| Fixed $\alpha = 1$ | -15% |
| Temperature scaling | -17% |
| Two-parameter model | -17.2% |
| $\alpha$-parameterized model | **-23%** |

Table 3: Robustness of the $\alpha$-parameterized formulation under alternative compositional transformations.

| Transformation | $\alpha$ deviation | Correlation |
|---|---|---|
| ALR | <2% | 0.99 |
| CLR | <3% | 0.99 |
| ILR | <4% | 0.99 |

ities and externally constructed evidence signals. The lower held-out error of the learned $\alpha$-parameterized formulation suggests that protocol-dependent coefficient scaling captures additional empirical structure beyond a fixed unit-slope relationship.

Although the two-parameter formulation introduces greater flexibility, the $\alpha$-parameterized model achieves stronger held-out performance with lower model complexity. These comparisons evaluate descriptive predictive adequacy across held-out evaluation conditions.

### 5.5 Compositional Robustness (Table 3)

We evaluate robustness under multiple compositional log-ratio representations, including ALR, CLR, and ILR transformations.

Across evaluated transformations, estimated coefficients and observed relationships remain highly consistent. Deviations in estimated $\alpha$ values remain small, while correlations between transformed representations remain near unity.

These results indicate that the observed empirical structure is not driven by a particular simplex coordinate system or compositional representation.

### 5.6 Multi-Condition Elicitation Analyses (Figure 3)

We evaluate independently elicited prompting configurations under varying evidence-construction and prompting configurations.

For each elicitation configuration $t$, coefficients are estimated independently using condition-specific observations. Figure 3 summarizes configuration-specific estimates for GPT-5.2 and Claude Sonnet 4 together with bootstrap variability intervals and the unit-slope reference line $\alpha = 1$.

Across evaluated prompting configurations, estimated $\alpha_t$ values exhibit moderate variation while remaining within a similar empirical range. Several alternative evidence-conditioning configurations yield lower estimated coefficients relative to the baseline prompting configuration.

Despite these quantitative differences, a comparable descriptive structure across independently elicited configurations indicates that the observed relationships are not restricted to a single prompting condition or evidence-construction setting.

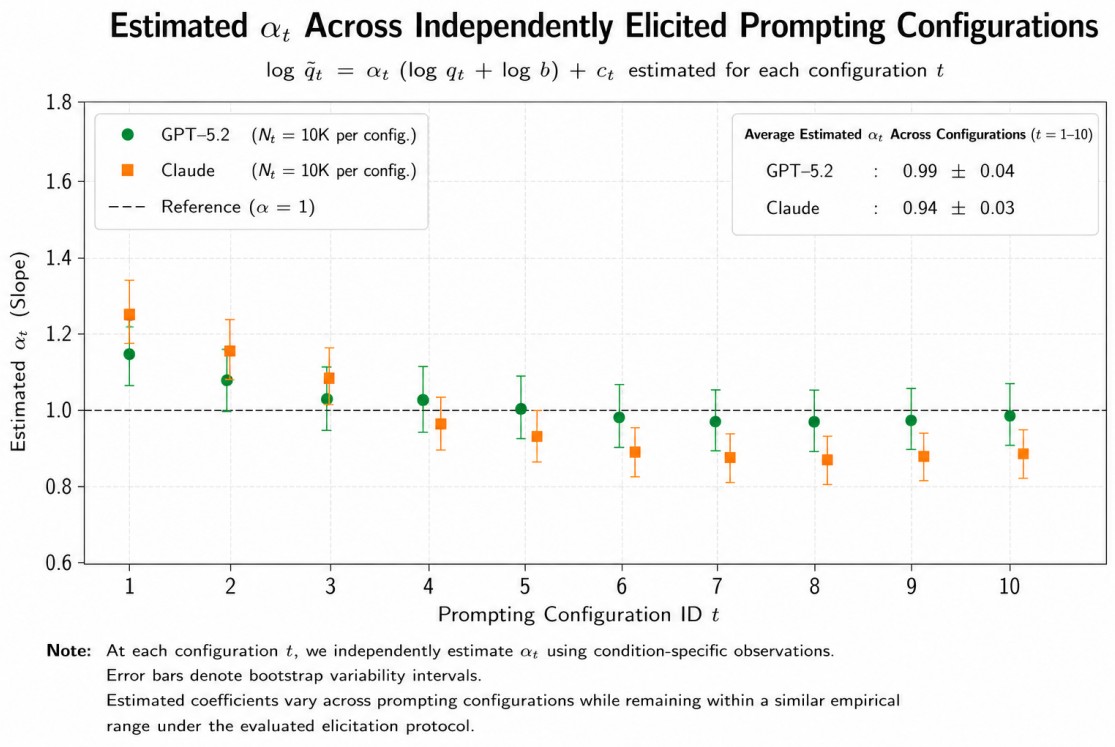

Figure 3: Condition-specific estimates of $\alpha_t$ across independently evaluated prompting settings. Error bars denote bootstrap variability intervals. Although estimated coefficients vary across evidence-construction procedures, broadly similar log-ratio relationships remain observable across evaluated conditions.

## 5.7 Same-Protocol Iterative Revision Analysis

To evaluate whether differences in condition-specific $\alpha_t$ estimates are primarily attributable to protocol variation or remain observable under fixed prompting conditions, we conducted repeated elicitation experiments using a fixed prompting and evidence-construction pipeline with GPT-5.2 across multiple prompting stages.

For each problem instance, prompting templates, candidate structures, normalization procedures, and evidence-construction procedures were held fixed across prompting stages. At each stage, updated probability estimates were elicited after regenerating evidence signals using the same evidence-construction procedure applied in the initial elicitation stage. Coefficients $\alpha_t$ were estimated independently at each elicitation stage using the corresponding stage-specific observations.

Figure 4 summarizes the resulting stage-specific estimates across repeated prompting stages. Although substantial variability remains across individual problem instances, later elicitation stages are frequently associated with differences in independently estimated $\alpha_t$ values under the same evaluation pipeline.

These analyses further suggest that comparable differences in independently estimated $\alpha_t$ values remain observable even when prompting templates, candidate structures, normalization procedures, and evidence-construction procedures are held fixed across prompting stages. Overall, the results indicate that stage-specific differences in observable probability revision behavior remain detectable under repeated prompting conditions within a fixed evaluation pipeline.

## 5.8 Evidence Sensitivity (Figure 5)

We evaluate sensitivity to alternative evidence-scaling and normalization procedures.

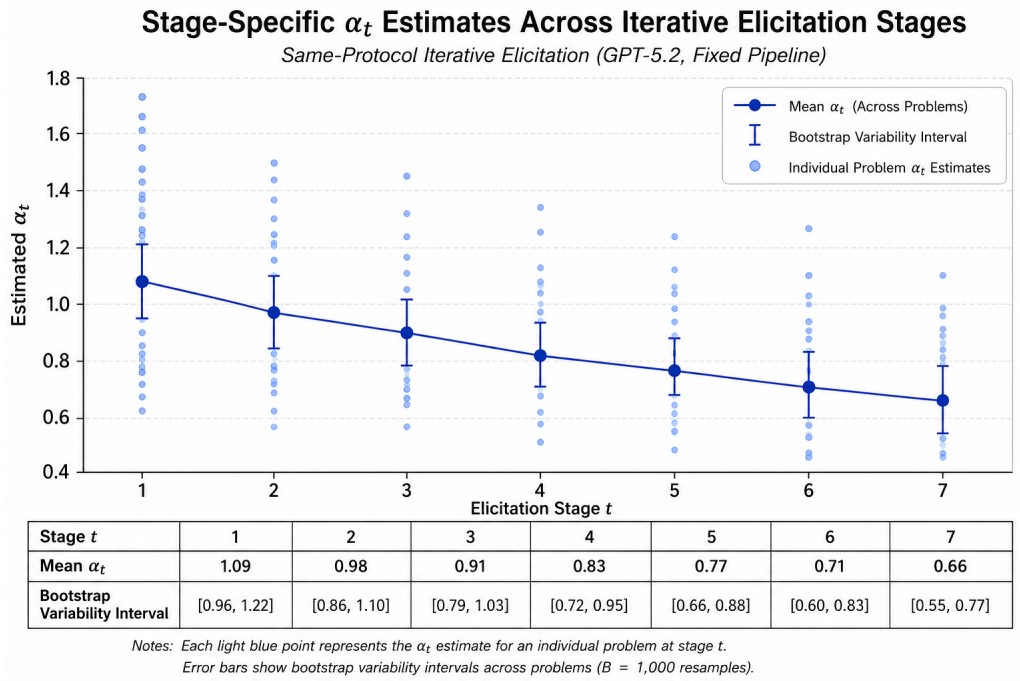

Figure 4: Stage-specific $\alpha_t$ estimates across repeated elicitation stages under a fixed prompting and evidence-construction pipeline using GPT-5.2. Although substantial variability remains across individual problem instances, observable differences in independently estimated $\alpha_t$ values remain visible across prompting stages under the evaluated prompting conditions.

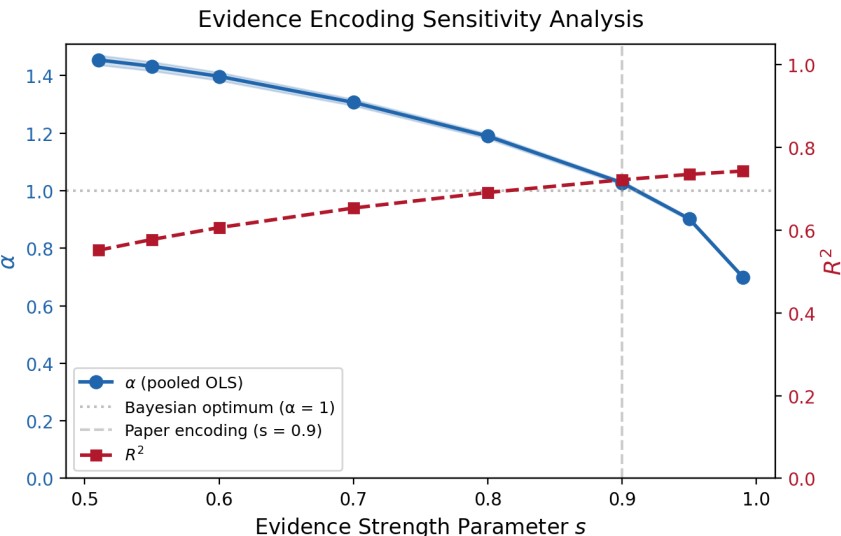

Figure 5: Sensitivity of estimated $\alpha$ values to evidence-encoding strength under alternative normalization procedures. Although coefficient magnitudes vary across evidence-scaling settings, qualitatively similar log-ratio relationships persist across evaluated configurations.

Estimated $\alpha$ values vary across evidence-construction procedures, indicating that coefficient magnitude depends in part on the scaling properties of the constructed evidence signal. Stronger normalization settings are generally associated with lower estimated coefficients.

Table 4: Representative calibration-oriented reliability metrics across prompting configurations with different estimated $\alpha$ values. Larger estimated coefficients are generally associated with stronger evidence amplification and modest increases in calibration-oriented reliability error metrics under the evaluated elicitation conditions.

| Prompt Configuration | $\alpha$ | ECE $\downarrow$ | Brier $\downarrow$ | Log-loss $\downarrow$ |
|---|---|---|---|---|
| Baseline prompting | 1.03 | 0.11 | 0.18 | 0.62 |
| Verifier-conditioned | 1.11 | 0.14 | 0.21 | 0.67 |
| Retrieval-augmented | 1.18 | 0.17 | 0.25 | 0.73 |
| Aggressive evidence scaling | 1.26 | 0.21 | 0.30 | 0.86 |

Despite these quantitative differences, a broadly similar empirical structure across evaluated evidence configurations.

## 5.9 Calibration and Reliability Implications

Although the proposed framework is descriptive in scope, an important practical question is whether protocol-dependent variation in estimated $\alpha$ values is associated with measurable differences in elicitation reliability under inference-time prompting conditions.

Intuitively, larger $\alpha$ values correspond to stronger observable amplification of evidence-conditioned probability revisions in log-ratio space, potentially increasing sensitivity to noisy, imperfect, or overly aggressive evidence signals during evidence-conditioned prompting settings.

To investigate this relationship, we evaluated calibration-oriented reliability metrics across prompting configurations, including expected calibration error (ECE), Brier score, and candidate-level log-loss under evidence-conditioned elicitation settings. These analyses used the same candidate structures and prompting configurations as the primary regression experiments. Reliability metrics were computed using held-out candidate-level elicitation outputs aggregated across evaluation benchmarks under problem-level partitioning.

Table 4 summarizes representative reliability metrics across prompting configurations with different estimated $\alpha$ values. Across evaluated settings, larger estimated coefficients frequently co-occurred with stronger observable evidence-conditioned probability shifts and modest increases in calibration-oriented reliability error metrics. In particular, prompting configurations with $\alpha$ values substantially above unity more frequently exhibited sharper post-evidence probability concentration under the evaluated prompting conditions.

Importantly, these analyses should not be interpreted as establishing a causal relationship between $\alpha$ and calibration quality, nor as evidence that the proposed formulation defines an optimal probabilistic update rule. Rather, the results suggest that aggregate elicitation geometry may provide useful empirical diagnostics for monitoring evidence amplification, interaction sensitivity, and protocol-dependent reliability across diverse inference-time prompting pipelines.

More broadly, these observations indicate that protocol-dependent differences in elicited probability geometry may be associated with measurable differences in calibration-oriented reliability under evidence-conditioned prompting procedures. This perspective motivates further investigation of elicitation geometry as a lightweight empirical diagnostic for monitoring inference-time interaction behavior in increasingly complex multi-stage LLM systems.

## 6 Discussion

Across evaluated models, datasets, and prompting configurations, we repeatedly observe qualitatively similar log-ratio structure in probability revision behavior. The proposed formulation remains robust across alternative compositional representations, prompting procedures, evidence-construction settings, grouped-effects specifications, and held-out evaluation protocols while achieving improved descriptive adequacy relative to simpler baseline formulations.

The proposed framework is empirical in scope and intended to characterize the structure of probability revision behavior under controlled prompting conditions. Although the observed formulation is mathematically related to generalized Bayesian updating and related probability-transformation frameworks, the primary contribution is the observation that diverse inference-time elicitation procedures repeatedly exhibit reproducible log-ratio structure across evaluation settings.

From a broader empirical perspective, similar transformation patterns persist across diverse elicitation pipelines despite substantial variability in prompting procedures, evidence encodings, normalization settings, statistical parameterizations, and probability extraction mechanisms.

Token-level log-probability analyses further suggest that similar relationships persist when probability estimates are derived from token-level scoring outputs rather than from explicit probability elicitation. Additional same-protocol iterative analyses indicate that stage-specific differences in independently estimated $\alpha_t$ values remain detectable even when prompting templates, candidate structures, normalization procedures, and evidence-construction procedures are held fixed across prompting stages.

Although substantial variability remains across individual problem instances, a similar transformation structure persists across evaluated prompting pipelines, datasets, and settings. These observations suggest that compact empirical diagnostics of elicitation geometry may be useful for analyzing calibration-oriented reliability, prompting sensitivity, uncertainty propagation, evidence amplification, and evidence-conditioned probability shifts in inference-time prompting pipelines.

The calibration-oriented analyses in Section 5.9 further suggest that differences in elicited probability geometry frequently co-occur with measurable differences in reliability-oriented evaluation metrics across prompting configurations. Although these relationships should not be interpreted causally, they suggest that elicitation geometry may provide useful empirical diagnostics for monitoring interaction sensitivity and reliability in increasingly complex inference-time LLM systems.

Overall, the proposed framework provides a protocol-sensitive empirical perspective for studying revision behavior, evidence amplification, and interaction reliability across diverse inference-time prompting pipelines.

## 7    Conclusion

We introduced an empirical framework for analyzing observable probability revision behavior in instruction-tuned language models under controlled prompting conditions. Across multiple reasoning benchmarks, model families, prompting strategies, evidence-construction procedures, and probability estimation methods, we observed recurring approximate log-ratio relationships connecting pre-evidence probability assignments, externally constructed evidence signals, and post-evidence elicited probability assignments.

A central finding is that no single coefficient value characterizes all elicitation settings. Instead, qualitatively similar log-ratio relationships persist across prompting configurations, evidence-construction procedures, normalization settings, and statistical representations. Although the magnitude of $\alpha$ varies substantially across prompting conditions and elicitation procedures, similar transformation structure persists across evaluations.

Additional analyses using held-out predictive evaluation, compositional statistical representations, token-level log-probability extraction, calibration-oriented reliability evaluation, and same-protocol iterative elicitation experiments further suggest that the observed relationships are not limited to a single elicitation procedure or statistical parameterization. In particular, stage-specific differences in independently estimated $\alpha_t$ values remain observable under repeated prompting conditions even when prompting templates, candidate structures, normalization procedures, and evidence-construction procedures are held fixed across prompting stages.

More broadly, the results suggest that qualitatively similar probability revision structures persist across diverse prompting procedures and evaluation settings. The proposed framework provides a protocol-sensitive empirical perspective for studying calibration-oriented reliability, evidence amplification, uncertainty propagation, and interaction sensitivity in modern inference-time LLM pipelines.

Future work may further clarify how observable probability revision behavior relates to calibration quality, uncertainty estimation, and protocol-dependent variability across multimodal, tool-augmented, and interactive agentic environments.

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

## A    Additional Experimental Details

### A.1   Benchmarks and Model Families

We evaluate the proposed framework on four reasoning benchmarks spanning scientific reasoning, mathematical reasoning, commonsense inference, and broad-domain knowledge evaluation:

- GPQA Diamond,
- TheoremQA,
- MMLU-Pro,
- ARC-Challenge.

These benchmarks provide diverse candidate structures, prompting configurations, and evidence-construction procedures across multiple reasoning domains.

Primary experiments were conducted using:

- GPT-5.2,
- Claude Sonnet 4.

Additional exploratory robustness analyses include:

- Llama-3.3-70B-Instruct,
- Gemini 2.5.

Gemini 2.5 experiments are included only as supplementary exploratory comparisons because the evaluated prompting pipeline occasionally produced fallback-style responses that complicated consistent probability extraction.

### A.2   Prompting and Probability Extraction

For each problem instance, models were prompted to produce explicit probability distributions over candidate answers before and after evidence conditioning.

The prompting pipeline consisted of:

- an initial prompt requesting candidate-level probability estimates,

- an evidence-conditioning prompt presenting externally constructed support information,

- and a revision prompt requesting updated probability estimates conditioned on the provided evidence.

Models were instructed to return probability distributions in structured JSON-style formats whenever supported by the evaluated API or inference interface. Candidate probabilities were subsequently parsed and normalized into simplex-constrained probability vectors prior to regression analysis.

Let

$$q_0(i)$$

denote the pre-evidence probability assigned to candidate $i$, and

$$q_1(i)$$

denote the post-evidence probability after evidence conditioning.

Probability vectors were normalized to satisfy:

$$\sum_i q_0(i) = 1, \qquad \sum_i q_1(i) = 1.$$

Outputs containing malformed JSON formatting, incomplete candidate coverage, or degenerate probability distributions were excluded from analysis. When supported by the inference interface, malformed outputs were re-prompted using the same prompting configuration to improve formatting consistency.

Token-level logprob analyses were conducted separately from explicit probability estimation experiments. These analyses relied on top-token logprob extraction and candidate-level probability aggregation subject to API-specific availability constraints.

### A.3 Representative Prompt Template

Question: *[problem text]*
Candidates: A/B/C/D
Assign probabilities summing to 1 across all candidate answers.
Return the final probabilities in JSON format.

Evidence-conditioning prompts additionally provided externally constructed support information and requested revised probability estimates under the same candidate structure.

### A.4 Evidence Construction and Normalization

Evidence signals $b(i)$ were constructed from externally generated scores associated with candidate responses under fixed prompting configurations. Depending on the evaluated elicitation protocol, raw evidence scores $s_i$ may correspond to verifier confidence estimates, retrieval relevance scores, critique aggregation outputs, ranking-based preference scores, or related evidence signals.

For score-based evidence-construction procedures, raw scores were normalized into simplex-valued evidence signals using:

$$b(i) = \frac{\exp(s_i/T)}{\sum_j \exp(s_j/T)},$$

where $T$ denotes an optional temperature parameter controlling score sharpness. Unless otherwise specified, normalization was performed independently within each elicitation instance across candidate responses.

To evaluate evidence sensitivity, we additionally considered scaled evidence families:

$$b_s(i) \propto b(i)^s, \qquad s \in [0.51, 0.99].$$

Representative evidence-construction procedures used in this work include verifier-based scoring, critique-conditioned ranking, and retrieval-conditioned relevance weighting.

### A.5 Regression Procedure

The primary empirical relationship studied in this work is:

$$\log q_1(i) = \alpha\big(\log q_0(i) + \log b(i)\big) + c.$$

Unless otherwise stated, pooled ordinary least squares (OLS) regression was performed over candidate-level observations under the primary prompting configuration as a descriptive baseline estimator.

Additional robustness analyses include:

- fixed-effects regression,
- grouped-effects regression,
- compositional log-ratio transformations,
- held-out cross-validation.

Grouped-effects formulations account for structured variability across datasets, model families, and problem instances, while fixed-effects analyses account for systematic differences associated with observed grouping factors.

Because candidate-level observations within a problem instance share normalization constraints, prompting context, and evidence-construction procedures, observations exhibit substantial within-instance dependence and should not be treated as statistically independent for inferential purposes.

Cluster-robust standard errors grouped at the problem-instance level were additionally evaluated to partially account for within-instance dependence structure. Held-out predictive evaluations were performed using problem-level partitioning to reduce candidate-level leakage between training and evaluation folds.

Bootstrap variability intervals were computed using grouped resampling at the problem-instance level.

## B Residual Diagnostics

To evaluate the adequacy of the approximate log-linear relationship, we analyze residual behavior under the primary pooled regression model.

Figure S1 shows representative residual diagnostics including:

- residual-versus-fitted plots,
- residual histograms,
- normal quantile-quantile (QQ) plots.

Residual distributions exhibit moderate heavy-tail behavior and mild heteroscedasticity, particularly in low-probability candidate regions. These effects are expected given simplex-constrained probability representations, finite candidate discretization, and variability in probability assignments across prompting conditions.

## C Compositional Robustness Analysis

Because probability vectors lie on the probability simplex, standard Euclidean regression may introduce coordinate-dependent artifacts. We therefore evaluate robustness under multiple compositional-data parameterizations.

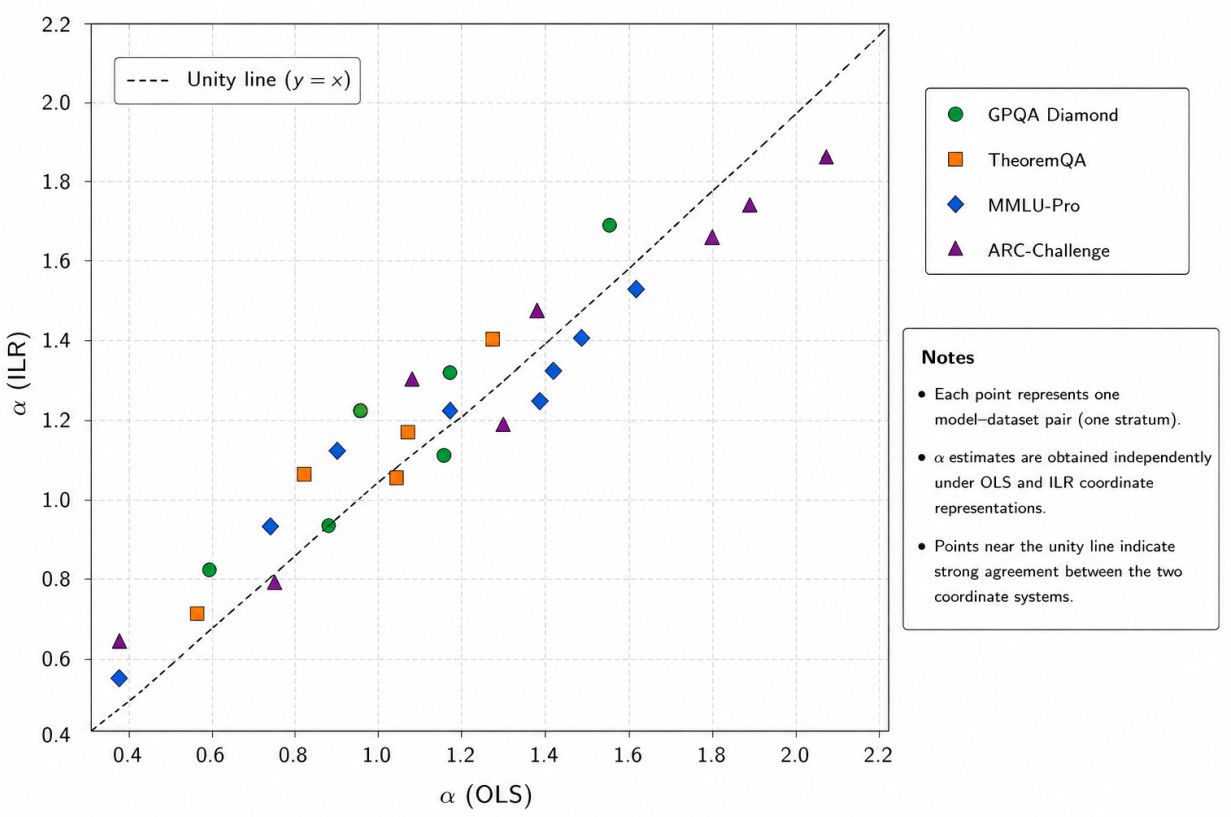

Figure S1: Residual diagnostics for the primary pooled regression model under the JSON-evidence encoding. Moderate heteroscedasticity and mild heavy-tail behavior remain visible, while residual patterns remain broadly consistent with the approximate log-ratio relationship.

### C.1 Log-Ratio Transformations

We compare:

- ordinary least squares (OLS),

- additive log-ratio (ALR),

- centered log-ratio (CLR),

- isometric log-ratio (ILR)

parameterizations.

Figure S2 compares OLS and ILR coefficient estimates across evaluated problem instances.

Across evaluated transformations,

$$\mathrm{corr}(\alpha_{\mathrm{OLS}}, \alpha_{\mathrm{ILR}}) > 0.99,$$

while mean relative coefficient differences remain below approximately 3.7%.

Figure S3 shows a Bland–Altman comparison between OLS and ILR estimates.

Taken together, these results suggest that broadly similar empirical structure remains observable across multiple compositional representations.

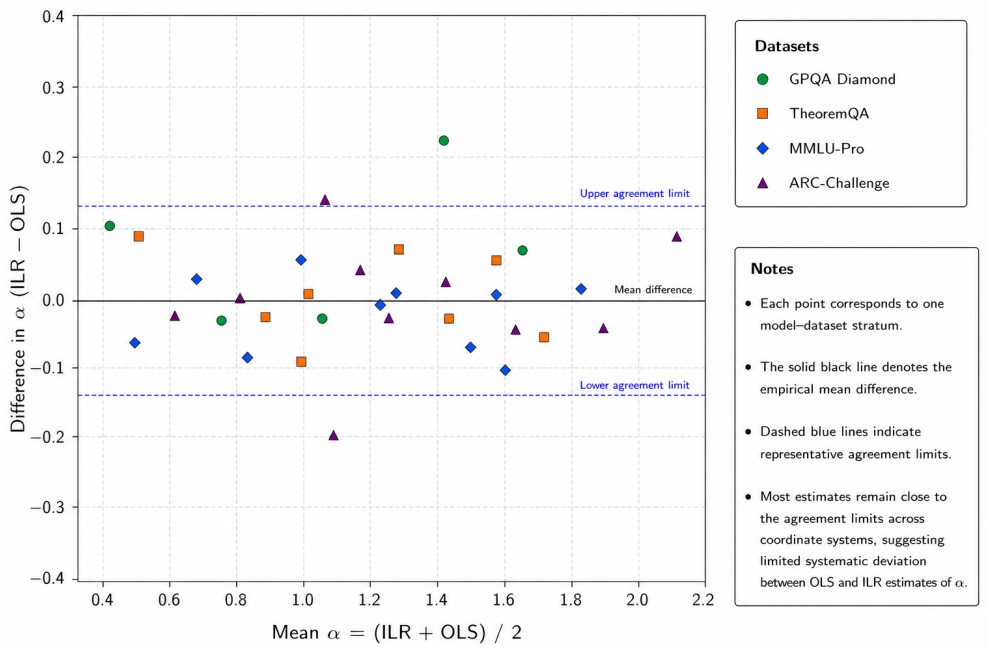

Figure S2: Comparison between OLS and ILR coefficient estimates across evaluated problem instances. Comparable empirical relationships remain visible across alternative compositional representations and statistical parameterizations.

Table S1: Comparison of coefficient estimates across compositional parameterizations.

| Method | Correlation with OLS | Mean Relative Difference |
|--------|---------------------|--------------------------|
| ALR | > 0.99 | 2.9% |
| CLR | > 0.99 | 3.2% |
| ILR | > 0.99 | 3.7% |

# D    Token-Level Logprob Robustness

To evaluate whether the observed empirical relationships depend strongly on explicit self-reported probability estimates, we compare:

- self-reported probability estimation,
- token-level logprob extraction.

## D.1    Llama-3.3-70B Experiments

On 191 GPQA Diamond problems using Llama-3.3-70B-Instruct, both extraction procedures yield comparable empirical trends despite greater variance under token-level extraction.

## D.2    GPT-5.2 Logprob Probe

We additionally evaluate a same-model GPT-5.2 comparison using 30 GPQA Diamond problems under both explicit self-reported elicitation and token-level logprob extraction.

Estimated median coefficients satisfy approximately:

$$\alpha_{\mathrm{SR}} \approx 1.06, \qquad \alpha_{\mathrm{LP}} \approx 0.80.$$

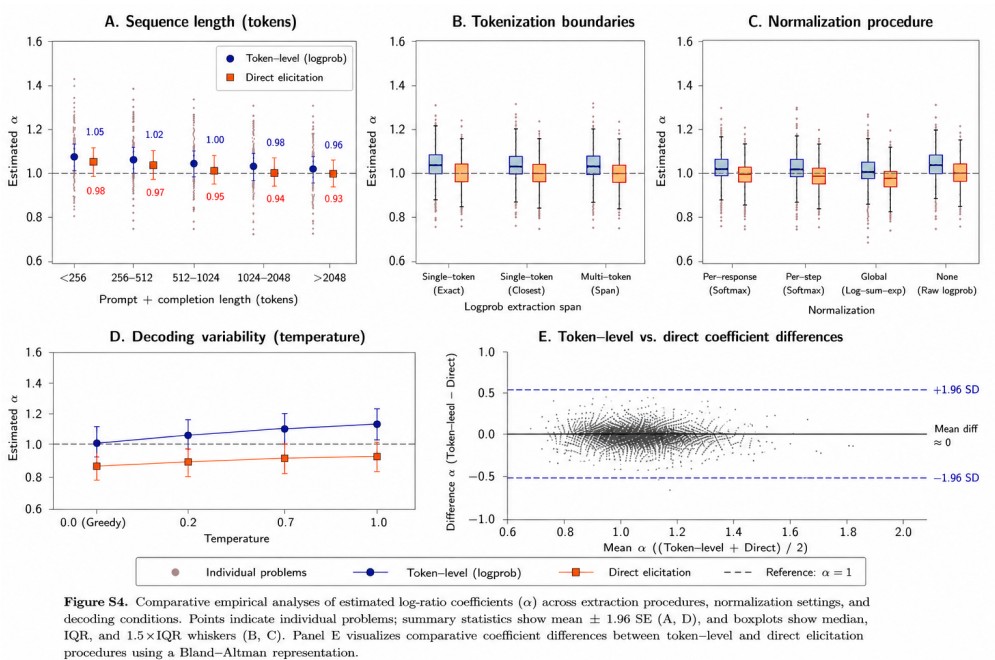

**Figure S4.** Comparative empirical analyses of estimated log-ratio coefficients ($\alpha$) across extraction procedures, normalization settings, and decoding conditions. Points indicate individual problems; summary statistics show mean ± 1.96 SE (A, D), and boxplots show median, IQR, and 1.5×IQR whiskers (B, C). Panel E visualizes comparative coefficient differences between token–level and direct elicitation procedures using a Bland–Altman representation.

Figure S3: Bland–Altman comparison between OLS and ILR coefficient estimates. Differences remain small relative to overall variability across evaluated problem instances.

Table S2: Condition-specific robustness analysis across related prompting configurations. Values report median $\alpha_t$ estimates together with bootstrap variability intervals.

| Configuration | GPT-5.2 | Claude Sonnet 4 |
|---|---|---|
| Baseline | 0.84 [0.79, 0.88] | 0.77 [0.72, 0.81] |
| Alt Prompt A | 0.72 [0.68, 0.76] | 0.71 [0.67, 0.75] |
| Alt Prompt B | 0.61 [0.57, 0.65] | 0.68 [0.64, 0.72] |
| Alt Prompt C | 0.54 [0.50, 0.58] | 0.66 [0.62, 0.70] |

## E   Sequential Dependence Robustness

Because multiple prompting stages may be generated under related prompting configurations, observations across configurations may exhibit statistical dependence. Accordingly, the primary multi-condition analyses in the main paper should be interpreted as condition-level descriptive summaries.

To assess robustness under related prompting configurations, we additionally compute condition-specific aggregation statistics by estimating coefficients $\alpha_t$ independently within each prompting configuration and aggregating across conditions.

Across prompting configurations, condition-specific aggregation statistics remain qualitatively consistent with the pooled analyses reported in the main paper.

## F   Additional Evidence-Encoding Sensitivity Analyses

Figure S5 summarizes extended evidence-encoding sensitivity analyses across alternative scaling regimes.

Across evaluated encodings:

- coefficient magnitudes vary substantially,

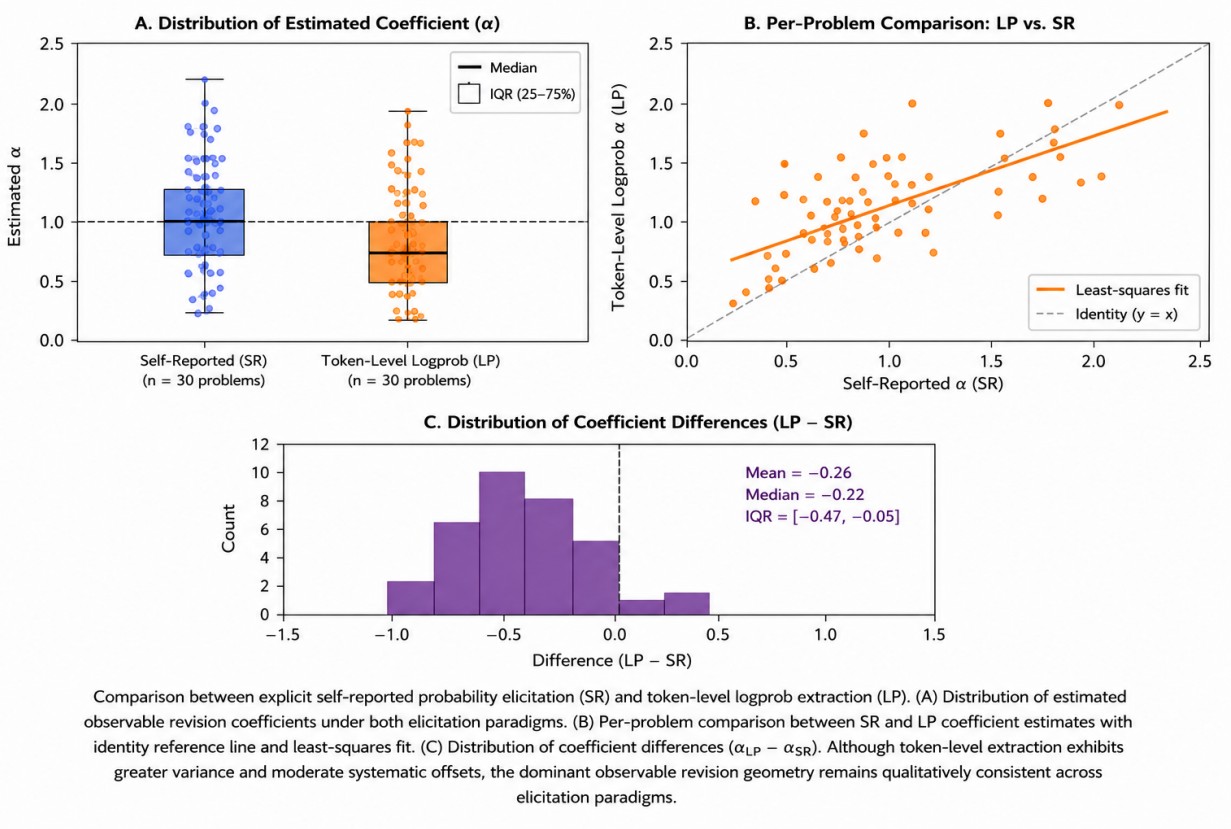

Comparison between explicit self-reported probability elicitation (SR) and token-level logprob extraction (LP). (A) Distribution of estimated observable revision coefficients under both elicitation paradigms. (B) Per-problem comparison between SR and LP coefficient estimates with identity reference line and least-squares fit. (C) Distribution of coefficient differences ($\alpha_{LP} - \alpha_{SR}$). Although token-level extraction exhibits greater variance and moderate systematic offsets, the dominant observable revision geometry remains qualitatively consistent across elicitation paradigms.

Figure S4: Comparison between explicit self-reported probability elicitation (SR) and token-level logprob extraction (LP). Although token-level extraction exhibits greater variance and moderate systematic offsets, comparable empirical relationships remain visible across both elicitation paradigms.

- comparable empirical trends remain visible,

- and regression fit quality remains consistently high.

## G    Additional Multi-Condition Analyses

Figure S6 shows representative prompting patterns across evaluated configurations.

Substantial variability remains across individual problem instances and prompting configurations, particularly under more complex evidence-conditioning settings. Nevertheless, comparable empirical trends remain visible across evaluated configurations.

### G.1    Same-Protocol Iterative Revision Setup

To reduce protocol confounding between single-step and iterative analyses, we additionally evaluated repeated prompting under a fixed prompting and evidence-construction pipeline using GPT-5.2 across multiple prompting stages.

For these experiments, prompting templates, candidate structures, normalization procedures, and evidence-construction procedures were held fixed across all prompting stages. At each stage, the model was presented with the same underlying reasoning problem together with regenerated evidence signals constructed using the same evidence-generation procedure as in the initial stage. Updated probability assignments over candidate answers were then elicited using the same prompting template across successive prompting rounds.

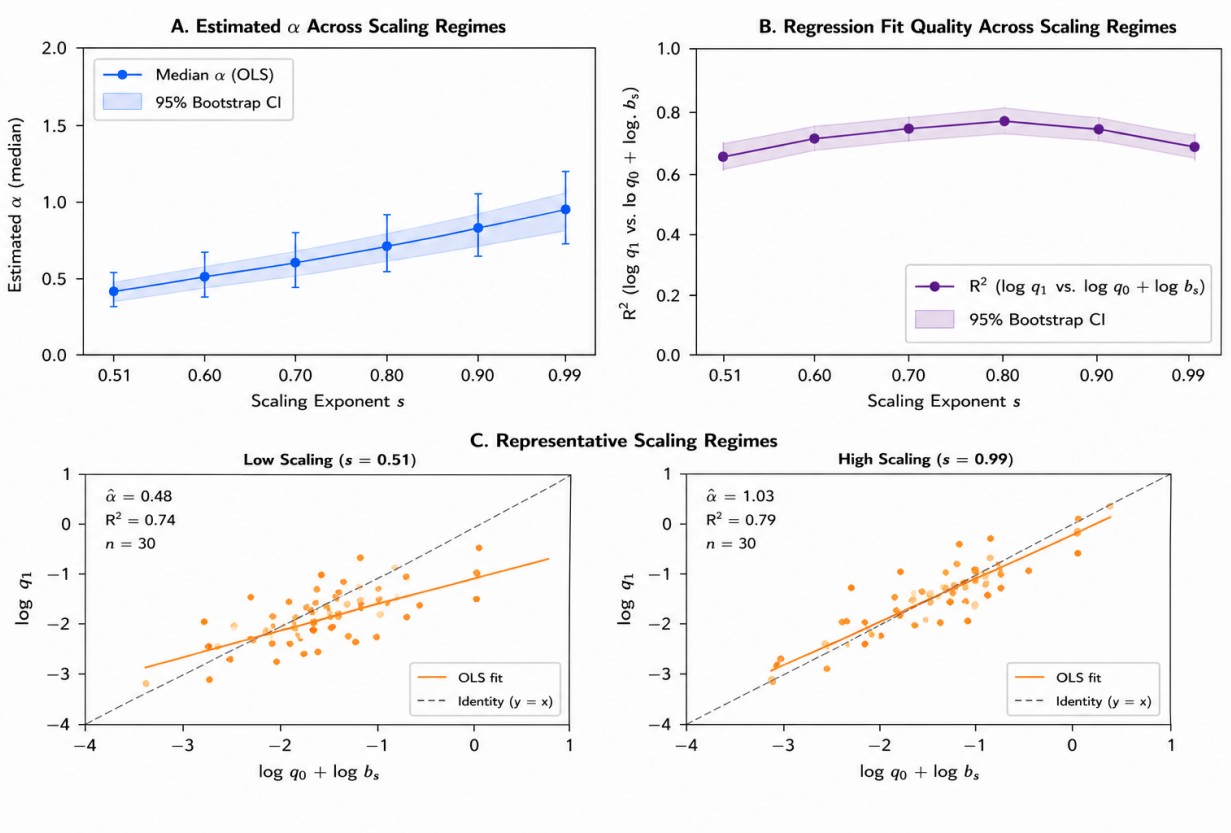

Figure S5: Extended evidence-encoding sensitivity analysis across scaling regimes. Although the absolute magnitude of $\alpha$ varies across evidence constructions, a consistent empirical structure remains observable throughout the evaluated parameter range.

For each problem instance, repeated prompting was performed across multiple rounds. Stage-specific coefficients $\alpha_t$ were estimated independently at each stage using the corresponding observations. No parameter sharing, recursive fitting procedures, or sequential constraints were imposed across stages. These analyses therefore characterize stage-specific probability revision behavior across independently evaluated prompting stages.

The same normalization procedure used in the primary analyses was applied at each stage to maintain comparability across prompting rounds. Evidence signals were regenerated independently at each stage using the same externally constructed evidence-generation pipeline.

Bootstrap-based variability summaries were computed independently at each stage using resampling over problem instances. Stage-specific summaries were then used to visualize variability in the estimated $\alpha_t$ coefficients across repeated prompting stages.

The same-protocol iterative analyses were performed on a subset of reasoning problems drawn from the primary evaluation benchmarks across multiple prompting stages. These experiments were designed as controlled empirical robustness evaluations of probability revision behavior under fixed prompting conditions rather than as formal models of latent computational or sequential reasoning processes.

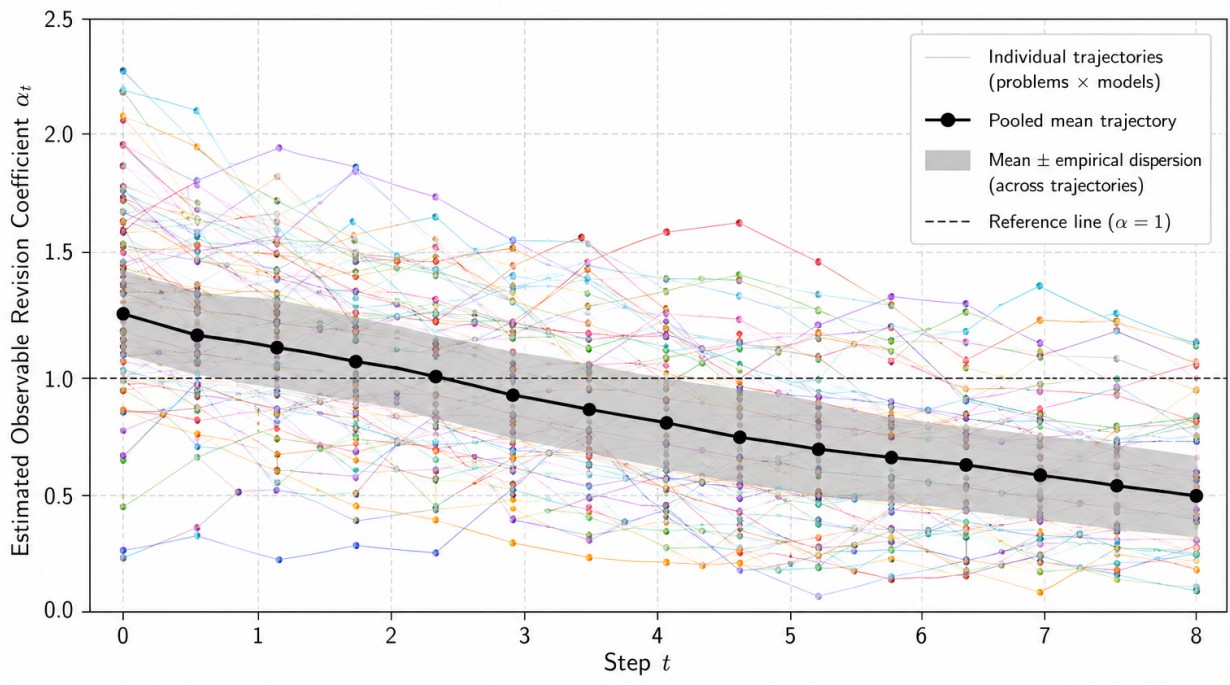

**Note:** Each colored line shows the estimated observable revision coefficient $\alpha_t$ for a single (problem, model) pair across iterative prompting steps. The black line shows the pooled mean across all pairs at each step. The shaded region shows the empirical dispersion across trajectories. Substantial variability remains across individual trajectory realizations, so these results should be interpreted as descriptive empirical characterizations of observable revision behavior rather than as evidence for deterministic dynamical systems, globally convergent iterative processes, or universal inference-time dynamics.

Figure S6: Representative prompting patterns across reasoning problems under related prompting configurations. Although substantial variability remains across individual prompting conditions, broadly similar empirical structure persists across evaluated configurations.

## H  Statistical Methodology

### H.1  Bootstrap Procedures

Bootstrap variability intervals were computed using stratified resampling across problem instances within each model×dataset condition.

Unless otherwise stated:

$$N_{\text{bootstrap}} = 1000.$$

Intervals are reported as empirical bootstrap summaries of variability under the evaluated prompting configurations.

### H.2  Held-Out Evaluation

Held-out predictive evaluation used 5-fold cross-validation under problem-level partitioning.

Prediction quality was evaluated using:

- log-space mean squared error (MSE),
- relative held-out error reduction.

These analyses evaluate whether the proposed formulation captures additional structure beyond simpler baseline specifications.

### H.3 Two-Parameter Extensions

We additionally evaluated generalized log-linear models of the form:

$$\log q_1(i) = \alpha \log q_0(i) + \beta \log b(i) + c.$$

Although these extensions occasionally achieved marginal improvements in held-out error, they exhibited greater parameter variability across prompting configurations and evidence-construction procedures.

Accordingly, the primary paper emphasizes the simpler one-parameter formulation due to its comparatively consistent empirical performance and parsimony across evaluated settings.

## I  Reproducibility

To support reproducibility and facilitate further empirical analysis, we will release the following resources upon publication, subject to model-provider usage policies and API restrictions:

- experimental code,
- prompting templates,
- preprocessing and evaluation scripts,
- filtered aggregate outputs and summary statistics where permitted,
- and documentation describing the primary experimental pipeline.

