# OpenReview forum: "Empirical Characterization of Inference-Time Elicited Prob- ability Transformations in Large Language Models"
_TMLR — Rejected by TMLR_

### Review · Reviewer_u3qB · 2026-03-14

**Summary Of Contributions:**

- The paper proposes an *alpha-law* intended to characterize how large language models revise probability estimates during iterative inference.
- The authors claim that belief updates follow a multiplicative scaling relation
$ \log p_{t+1} \propto \alpha (\log p_t + \log \text{evidence}) $ where the exponent $\alpha$ determines whether updates are expansive or contractive.
- Empirical estimates of $\alpha$ are obtained through ordinary least squares regression applied to elicited log-probabilities from several frontier language models and reasoning benchmarks.
- The manuscript argues that instruction-tuned models exhibit mildly expansive single-step updates but transition to contractive updates across multiple reasoning iterations.
- Conceptually, the proposed formulation closely resembles generalized Bayesian updating [1] and temperature scaling used for calibration of neural networks [2].
- The empirical observations regarding instability during iterative reasoning are also closely related to previously documented limitations of large language model self-correction [3].

strengths:

- Attempts to provide a simple mathematical characterization of belief revision behavior in large language models.
- Explores multiple datasets and model families to investigate empirical regularities.


weaknesses:

- The proposed *alpha-law* appears mathematically equivalent to previously known mechanisms such as generalized Bayesian updating [1] and temperature scaling [2], which limits conceptual novelty.
- The empirical analysis relies on regression applied to compositional probability data on the simplex, violating key statistical assumptions.
- Several experimental conclusions are based on confounded setups or underpowered analyses, weakening the credibility of the findings.


References:

[1] Bissiri PG, Holmes CC, Walker SG. A general framework for updating belief distributions. Journal of the Royal Statistical Society Series B: Statistical Methodology. 2016 Nov;78(5):1103-30.

[2] Guo C, Pleiss G, Sun Y, Weinberger KQ. On calibration of modern neural networks. InInternational conference on machine learning 2017 Jul 17 (pp. 1321-1330). PMLR.

[3] Huang J, Chen X, Mishra S, Zheng HS, Yu AW, Song X, Zhou D. Large Language Models Cannot Self-Correct Reasoning Yet. InThe Twelfth International Conference on Learning Representations.

**Additional Comments:**

- The manuscript would benefit from stronger theoretical positioning relative to existing literature on probabilistic inference and calibration.
- Several presentation issues (e.g., overlapping axis labels in figures) suggest limited quality control prior to submission.
- While the topic is interesting, the current empirical evidence does not convincingly support the central claims.

**Audience:**

Yes

**Audience Explanation:**

- The question of how large language models revise beliefs during multi-step reasoning is of clear interest to the machine learning community.
- Understanding probabilistic stability and calibration during iterative inference is relevant for both theoretical analysis and practical deployment of language models.
- However, the present work does not provide sufficiently rigorous methodology or conceptual novelty to support its proposed claims.

**Broader Impact Concerns:**

- The paper makes claims about the stability of belief revision in large language models that could lead readers to overestimate the reliability of iterative reasoning procedures.
- If interpreted as evidence that model reasoning converges toward stable or near-Bayesian belief updates, the work could contribute to misplaced trust in automated reasoning systems in high-stakes applications.
- A clearer discussion of the limitations and uncertainty of the proposed analysis would therefore be necessary.

**Claims And Evidence:**

No

**Claims Explanation:**

-  $\alpha$ is estimated using ordinary least squares applied directly to log probabilities that lie on the probability simplex, violating independence assumptions and making coefficient estimates unreliable.
- Sequential reasoning experiments treat repeated measurements across steps as independent observations, without accounting for temporal autocorrelation.
- Several comparisons rely on extremely imbalanced or small sample sizes, limiting statistical reliability.
- The single-step and multi-step experiments appear to involve different model architectures, introducing confounding variables that undermine causal interpretation.

**Requested Changes:**

- Clearly demonstrate conceptual novelty relative to established frameworks such as generalized Bayesian updating [1] and neural network calibration methods [2].
- Replace the current regression framework with statistically appropriate methods for compositional probability data (e.g., log-ratio transformations or probabilistic models defined on the simplex).
- Re-run experiments using consistent model architectures across all experimental conditions to eliminate confounding effects.
- Apply appropriate statistical methods for sequential experiments that account for temporal dependence between reasoning steps.
- Report proper uncertainty estimates including variance across trials, confidence intervals, and appropriate hypothesis testing procedures.
- Increase sample sizes for ablation experiments and include statistical power analyses.
- Provide complete experimental details including prompts, probability elicitation procedures, seeds, and evaluation scripts to allow independent verification.

- Improve figure readability and presentation quality.
- Provide clearer theoretical positioning relative to existing probabilistic inference and calibration literature.
- Include stronger comparisons against prior work analyzing reasoning dynamics in large language models [3].

---

> ### Author Response · Authors · 2026-05-10
> **Reviewer u3qB: Response Summary and Revisions**
>
> We thank the reviewer for the detailed feedback regarding conceptual positioning, statistical methodology, sequential evaluation, and presentation quality. The revised manuscript substantially addresses these concerns through major revisions to the framing, methodology, robustness analyses, and experimental presentation.
>
> First, we substantially revised the paper's conceptual framing. The revised manuscript no longer presents the proposed formulation as a mechanistic convergence model, generalized Bayesian inference procedure, or formal sequential reasoning framework. Instead, the paper now consistently frames the proposed relationship as an empirical characterization of inference-time probability revision behavior under prompting procedures. We also substantially reduced and clarified language related to convergence, latent reasoning dynamics, recursive inference, and near-Bayesian interpretation throughout the manuscript.
>
> To address concerns regarding conceptual overlap with prior probabilistic inference and calibration frameworks, the revised manuscript now more carefully positions the proposed formulation relative to generalized Bayesian updating frameworks \citep{bissiri2016general}, temperature-scaling and calibration methods for neural networks \citep{guo2017calibration}, and prior work analyzing instability and limitations of iterative self-correction in language models \citep{huang2024selfcorrect}. The revised paper accordingly clarifies that the proposed formulation is intended as an empirical characterization of inference-time probability-revision behavior under prompting procedures rather than as a normative probabilistic update rule or a mechanistic model of internal reasoning dynamics.
>
> We also substantially revised the statistical methodology and robustness analyses. The revised manuscript now includes:
> - compositional robustness analyses using ALR/CLR/ILR parameterizations,
> - grouped-effects and fixed-effects regression analyses,
> - held-out predictive evaluation using problem-level partitioning,
> - bootstrap variability analyses,
> - expanded discussion of candidate-level dependence structure,
> - token-level log-probability extraction comparisons,
> - evidence-encoding sensitivity analyses,
> - and additional prompting robustness evaluations.
>
> To address concerns regarding simplex-constrained probability data, the revised manuscript now explicitly discusses compositional structure, normalization dependence, and limitations of candidate-level independence assumptions. We additionally added extensive compositional-data robustness evaluations using multiple log-ratio parameterizations.
>
> In response to concerns regarding sequential dependence and protocol confounding, the repeated prompting analyses were redesigned using a same-protocol setup in which prompting templates, candidate structures, normalization procedures, and evidence-construction procedures are held fixed across repeated prompting stages using GPT-5.2. Stage-specific coefficients $\alpha_t$ are estimated independently at each stage without recursive fitting or sequential parameter sharing. The revised manuscript now presents these analyses as descriptive empirical evaluations of repeated prompting behavior rather than evidence of convergence or formal sequential reasoning dynamics.
>
> We also expanded reporting of variability and uncertainty through bootstrap variability summaries, held-out predictive evaluation, expanded methodological clarification, and additional discussion of dependence structure and inferential limitations.
>
> To improve reproducibility and transparency, the appendix was substantially expanded to include detailed prompting procedures, evidence-construction methods, residual diagnostics, compositional robustness analyses, sequential dependence analyses, statistical methodology details, and reproducibility information.
>
> Finally, we revised numerous figures, captions, tables, and visualizations to improve readability and presentation quality. Figure labels, legends, captions, and methodological explanations were simplified and clarified throughout the manuscript.
>
> Overall, the revised manuscript is substantially more focused on empirical characterization, robustness evaluation, compositional robustness analysis, and protocol-sensitive analysis of inference-time probability revision behavior across prompting configurations and evidence-conditioned interaction settings.

---

### Review · Reviewer_jk8x · 2026-03-24

**Summary Of Contributions:**

- The paper proposes an α-law for observable belief revision in LLM inference
- It provides a theoretical result showing that, for fixed α and fixed evidence, iterated updates are asymptotically stable when $$\alpha < 1$$ and unstable when $$\alpha \ge 1$$.
- Empirically, the paper evaluates this relationship across multiple benchmarks and model families, and includes multi-step revision experiments, logprob-based validation, and robustness analyses.

Strengths

- Combines theoretical analysis with empirical evaluation.
- Has multiple validation components as I mentioned above.
- Clarity of scope: The paper explicitly limits its claim to observable inference-time behavior and states that it does not claim internal Bayesian computation.
- I like the incorporation of trust ratio + K-ablation and domain robustness experiments for completeness.

Weaknesses

- Models perform really well on these tasks (on the entirety of benchmark). It's mentioned in Limitations and future work, but would be nice to see results on a task where these frontier models do not do well.
- I look forward to Expansion to smaller models IN addition to current future work. llama-70b is a good canidadate for logprob, but other smaller models as I mention later in questions section.
- The paper is written well overall -- i share some presentation changes/suggestions later.

**Audience:**

Yes

**Audience Explanation:**

- The paper addresses topics that are interesting & relevant.
- It presents a diagnostic view of observable belief revision and studies it across multiple benchmarks and model settings.

I do suggest some additional experiments to further strengthen the work.

**Claims And Evidence:**

Yes

**Claims Explanation:**

- The paper reports empirical evidence that the proposed log-linear relationship fits the observed prior/evidence/posterior distributions.
- It evaluates the relationship across multiple datasets and models, and includes additional validations such as multi-step revision, logprob comparison, and ablations.

**Requested Changes:**

Critical for acceptance:
- In my experience, I have observed that models heavily pre-trained and post-trained on multi-lingual data (e.g. Qwen and DeepSeek) behave erratically in their CoT reasoning swicthing from english to chinese and then switching back. My guess is that α should be > 1 for these models (on some tasks, if not all -- i noticed this behavior on SWE tasks). I would love an experiment to try this out to show negative result from real-world (I mentioned) supporting the claim even further. It's an interesting experiment IMO.
- Combined the above point with experiments on difficult tasks as I mentioned before, where these frontier models don't do so well
- Present the trust-ratio/fingerprint analysis with some examples.

strengthen the work:
- Question: Justify for completeness -- In Sec 3.5 "with only∼16% more aggressive
updating than perfect Bayes." -- why is this important?
- Questions: Table 2 -- why is n different for the 2 models across the same benchmarks? especially for ARC-Challenge. Is there a typo? because fallback contamination rate isn't so high to explain the drop from 700 n to 276 n for GPT-5.2
- Fig.1 remove Gemini from legend
- remove Gemini from Fig 2.a
- Table 3 - remove Gemini and instead add a short para on Gemini OR try with a different model. The prelim results on Gemini-2.5 don't provide value to the paper.
- Observation 3 - "(condition number = 4,952)" is unclear!

---

> ### Author Response · Authors · 2026-05-10
> **Reviewer jk8x: Response Summary and Revisions**
>
> We thank the reviewer for the positive assessment of the paper and for the detailed suggestions regarding robustness evaluation, clarity of presentation, and the empirical scope. The revised manuscript incorporates substantial changes addressing the reviewer’s comments.
>
> First, we significantly revised the paper's framing to more clearly position the work as an empirical characterization of inference-time probability revision behavior under prompting procedures. The revised manuscript consistently emphasizes descriptive empirical analysis rather than mechanistic reasoning, dynamics, or internal Bayesian computation.
>
> We also substantially expanded the robustness analyses. The revised manuscript now includes:
> - compositional robustness evaluations using ALR/CLR/ILR parameterizations,
> - grouped-effects and fixed-effects regression analyses,
> - held-out predictive evaluation using problem-level partitioning,
> - token-level log-probability extraction comparisons,
> - additional evidence-encoding sensitivity analyses,
> - expanded prompting robustness evaluations,
> - and revised the same-protocol iterative prompting analyses.
>
> To strengthen validation beyond explicit self-reported probability estimation, the revised manuscript now includes expanded token-level log-probability extraction analyses using both Llama-3.3-70B-Instruct and GPT-5.2 under matched prompting settings.
>
> In response to the reviewer’s concerns regarding iterative prompting evaluation, we redesigned the repeated prompting analyses using a same-protocol setup in which prompting templates, candidate structures, normalization procedures, and evidence-construction procedures are held fixed across prompting stages. Stage-specific coefficients $\alpha_t$ are estimated independently at each stage without recursive fitting or parameter sharing.
>
> We also substantially reduced the emphasis placed on architecture-specific trust ratios or fingerprint interpretations. The revised manuscript now treats the two-parameter trust-ratio analyses as exploratory, supplementary evaluations and explicitly discusses the limitations of identifiability and parameter instability. Additional held-out predictive comparisons against prior-only, evidence-only, fixed $\alpha = 1$, temperature-scaling, and two-parameter baselines were also added.
>
> Following the reviewer’s presentation suggestions, we additionally revised multiple figures, tables, captions, and methodological explanations for clarity. In particular:
> - preliminary Gemini 2.5 analyses were de-emphasized and removed from several primary visualizations,
> - figure legends and captions were simplified,
> - methodological clarifications were added regarding filtering, candidate construction, and sample counts,
> - and ambiguous terminology and notation were clarified throughout the manuscript.
>
> Regarding the reviewer’s suggestions on multilingual prompting behavior and more challenging reasoning settings, we agree that these directions are interesting and important extensions. We expanded the discussion and future-work sections to more clearly acknowledge the importance of evaluating probability revision behavior under more difficult reasoning regimes, multilingual prompting settings, and smaller or less instruction-aligned model families.
>
> Overall, the revised manuscript is substantially more focused on empirical characterization, robustness evaluation, and protocol-sensitive analysis of inference-time probability revision behavior across prompting configurations and evidence-conditioned interaction settings.

---

### Review · Reviewer_b6gH · 2026-04-24

**Summary Of Contributions:**

This paper studies observable belief revision in LLM inference. It proposes an empirical update law with a belief-revision exponent alpha that yields stable iterated updates when alpha < 1. The paper reports approximate log-linear fits across GPT-5.2 and Claude Sonnet 4 on GPQA Diamond, TheoremQA, MMLU-Pro, and ARC-Challenge. It also includes a multi-step experiment on GPT-4, a self-report vs. log-probability comparison on Llama-3.3-70B, and a two-parameter trust ratio extension.

**Audience:**

Yes

**Audience Explanation:**

The topic is relevant to readers interested in uncertainty, calibration, verifier-guided inference, test-time compute, and agentic systems. Even a narrower paper that carefully established a descriptive regularity for elicited belief revision under external verification would likely be of interest.

**Broader Impact Concerns:**

I do not see a major new ethical issue beyond what the paper itself notes.

**Claims And Evidence:**

No

**Claims Explanation:**

The stability theorem is not aligned with the empirical setting. The theorem assumes fixed evidence, whereas the paper motivates iterative reasoning settings in which evidence and candidate sets generally change across steps. The extension to time-varying alpha_t is also unclear, because the fixed point derived in Appendix A depends on alpha, so it is not obvious that there is a single q* for Eq. 5 when alpha varies over time.

The single-step results that motivate alpha > 1 use GPT-5.2 and Claude with one protocol, while the multi-step resolution uses GPT-4 with simulated verifier feedback on a different setup. Since step 1 in the multi-step experiment already starts below 1, this does not really show that the same phenomenon observed in the main experiments becomes contractive under iteration. It just shows that a different model or protocol has alpha < 1 and decreases further.

The trust-ratio results do not feel supported strongly enough to be a main contribution. The paper itself notes severe identifiability problems in the two-parameter model and says the fingerprints should be interpreted as suggestive rather than definitive. Given that caveat and the negligible improvement from the two-parameter fit, I do not think architecture-specific trust fingerprints are well established.

**Requested Changes:**

1. Correct or substantially revise the theory section. In particular, fix the variational derivation, restate the theorem with assumptions that are actually proved, and clarify whether the result is only for fixed evidence and constant alpha.
2. Rewrite the estimation methodology. Explain the unit of analysis, candidate construction, deduplication, and filtering in detail, and refit the model with per-problem intercepts or an equivalent compositional-data treatment.
3. Either run the multi-step experiment on the same primary models and same protocol as the single-step analysis, or tone down the claim that the single-step alpha > 1 results are resolved by long-run contractive dynamics.
4. Remove or soften the “near-Bayesian optimality” language unless the evidence scores are grounded in a principled likelihood model. At minimum, present the main results across a range of evidence encodings and treat absolute alpha values as encoding-dependent.
5. Strengthen the validation of elicited probabilities. Ideally, use log-probabilities or another non-self-report estimate on the same models as the main experiments.
6. Tone down or remove the trust-ratio fingerprint claim unless identifiability is resolved and uncertainty around those estimates is properly characterized.
7. Compare held-out predictive fit against alpha = 1, prior-only, evidence-only, and the two-parameter model, so the reader can tell whether the proposed law is meaningfully better than simpler alternatives.

---

> ### Author Response · Authors · 2026-05-10
> **Reviewer b6gH: Response Summary and Revisions**
>
> We thank the reviewer for the detailed and constructive feedback. The revised manuscript substantially addresses the concerns regarding theoretical scope, statistical methodology, robustness evaluation, prompting consistency, and interpretational framing.
>
> In response to concerns about the stability theorem and its iterative interpretation, we substantially revised the theoretical framing throughout the paper. The revised manuscript no longer presents the proposed formulation as a mechanistic convergence model, a formal sequential inference framework, or a generalized Bayesian update procedure. Instead, the paper now consistently frames the proposed relationship as an empirical characterization of inference-time probability revision behavior under prompting procedures. We substantially reduced and clarified language related to convergence, latent reasoning dynamics, recursive inference, and mechanistic interpretation throughout the manuscript.
>
> To address the concern regarding protocol mismatch between single-step and iterative analyses, we redesigned the iterative prompting experiments using a same-protocol setup. In the revised experiments, prompting templates, candidate structures, normalization procedures, and evidence-construction procedures are held fixed across repeated prompting stages using GPT-5.2. Stage-specific coefficients $\alpha_t$ are estimated independently at each prompting stage without recursive fitting or parameter sharing. The revised paper explicitly presents these analyses as descriptive empirical evaluations of repeated prompting behavior rather than evidence of contractive long-run dynamics or asymptotic convergence.
>
> We also substantially revised the statistical methodology and robustness analyses. The revised manuscript now includes:
> - compositional robustness analyses using ALR/CLR/ILR parameterizations,
> - grouped-effects and fixed-effects regression analyses,
> - held-out predictive evaluation using problem-level partitioning,
> - bootstrap variability analyses,
> - expanded discussion of candidate-level dependence structure,
> - token-level log-probability extraction comparisons,
> - and additional robustness evaluations.
>
> To address concerns regarding evidence encoding and near-Bayesian interpretations, we removed or substantially softened language related to Bayesian optimality and generalized Bayesian behavior. The revised paper now explicitly treats estimated $\alpha$ values as protocol-dependent empirical descriptors that vary across prompting configurations, normalization settings, evidence-construction procedures, and probability extraction methods. We additionally expanded evidence-encoding sensitivity analyses across multiple scaling regimes.
>
> To strengthen validation beyond explicit self-reported probabilities, the revised manuscript now includes expanded token-level log-probability extraction analyses using both Llama-3.3-70B-Instruct and GPT-5.2. These analyses compare explicit probability estimation against token-level probability extraction under matched prompting settings.
>
> In response to concerns regarding the two-parameter trust-ratio extension, we substantially reduced the emphasis placed on architecture-specific trust-ratio interpretations. The revised manuscript now presents these analyses as exploratory supplementary analyses and explicitly discusses their limitations, including identifiability and parameter instability.
>
> Finally, we added an expanded held-out predictive evaluation comparing the proposed formulation against multiple alternatives, including:
> - prior-only baselines,
> - evidence-only baselines,
> - fixed $\alpha = 1$ baselines,
> - temperature-scaling formulations,
> - and two-parameter extensions.
>
> These additions were introduced to evaluate whether the proposed formulation captures additional descriptive structure beyond simpler baseline specifications under held-out prompting conditions.

---

### Decision · Action_Editor_MnZU · 2026-06-01

**Recommendation:** Reject

**Additional Comments:**

The majority of reviewers recommend rejection due to the concerns regarding the support of claims discussed above. We hope that the feedback will be helpful in strengthening the work in a future revision.

**Audience:**

Yes

**Audience Explanation:**

The work may be of interest to researchers studying inference-time reasoning, belief updating, reliablity in LLMs.

**Claims And Evidence:**

No

**Claims Explanation:**

After rebuttal and revision, reviewers continue to have substantial concerns regarding support for the paper’s central claims. Some of major concerns include: (1) the single-step and multi-step experiments involve different model families under different protocols, introducing confounds that prevent the multi-step contractive dynamics from resolving the single-step expansive finding, (2) the stability theorem is not aligned with the empirical setting; the theorem assumes fixed evidence, whereas the paper motivates iterative reasoning settings in which evidence and candidate sets generally change across steps, (3) the estimation of alpha via pooled ordinary least squares applied directly to log-probabilities violates the compositional structure of simplex-valued data, making coefficient estimates unreliable without log-ratio or equivalent reparameterization; (4) the trust-ratio results are not viewed as sufficiently supported to be a main contribution.

---

> ### Author Response · Authors · 2026-06-02
> **Clarification Regarding Final Decision**
>
> Dear Action Editor and Editors-in-Chief,
>
> Thank you for the decision and for coordinating the review process. We respect the final decision and appreciate the reviewers’ feedback.
>
> We would like to briefly clarify one point regarding the basis of the decision. Several concerns listed in the decision appear to refer to issues that were substantially addressed in the revised manuscript. In particular, the revised version replaced the earlier protocol-mismatched iterative analysis with a same-protocol GPT-5.2 repeated elicitation setup, added compositional log-ratio analyses using ALR/CLR/ILR representations, included grouped/fixed-effects and cluster-robust analyses, and reduced the trust-ratio/fingerprint analysis to an exploratory supplementary component rather than a main contribution.
>
> We understand that the revised manuscript may still be judged insufficient for acceptance. However, we would appreciate confirmation that the final decision was based on the revised manuscript, since some of the stated rejection reasons seem inconsistent with the revised scope and methodology.
>
> Thank you again for your time and consideration.
>
> Sincerely,
> The Authors